# Multi-tissue expression and splicing data prioritise anatomical subsite- and sex-specific colorectal cancer susceptibility genes

Emma Hazelwood [1,2], Daffodil M. Canson [3], Benedita Deslandes [1,2], Xuemin Wang[4], Pik Fang Kho[5], Danny Legge[6], Andrei-Emil Constantinescu[1,2], Matthew A. Lee [7], D. Timothy Bishop [8], Andrew T. Chan [9,10,11,12,13,14], Stephen B. Gruber[15], Jochen Hampe [16], Loic Le Marchand[17], Michael O. Woods[18], Rish K. Pai[19], Stephanie L. Schmit [20,21], Jane C. Figueiredo[22], Wei Zheng[23], Jeroen R. Huyghe [24], Neil Murphy [7], Marc J. Gunter[7,25], Tom G. Richardson [1,2], Vicki L. J. Whitehall [26,27,28], Emma E. Vincent [1,2,6], Dylan M. Glubb [4,29] & Tracy A. O'Mara [4,27,29] ✉

Genome-wide association studies have suggested numerous colorectal cancer (CRC) susceptibility genes, but their causality and therapeutic potential remain unclear. To prioritise causal associations between gene expression/splicing and CRC risk (52,775 cases; 45,940 controls), we perform a transcriptome-wide association study (TWAS) across six tissues with Mendelian randomisation and colocalisation, integrating sex- and anatomical subsite-specific analyses. Here we reveal 37 genes with robust causal links to CRC risk, ten of which have not previously been reported by TWAS. Most likely causal genes with evidence of cancer cell dependency show elevated expression linked to risk, suggesting therapeutic potential. Notably, *SEMA4D*, encoding a protein targeted by an investigational CRC therapy, emerges as a key risk gene. We also identify a female-specific association with CRC risk for *CCM2* expression and subsite-specific associations, including *LAMC1* with rectal cancer risk. These findings offer valuable insights into CRC molecular mechanisms and support promising therapeutic avenues.

Colorectal cancer (CRC) is the third most common cancer worldwide and the fourth most common cause of cancer-related death[1]. There are several established risk factors for CRC, including obesity, alcohol consumption and tobacco use[2–9] and there is evidence of heterogeneity by sex and anatomical site[2,10]. However, the biological pathways that causally affect CRC development remain poorly understood, which has limited the ability to design suitable therapeutic interventions for prevention and treatment[2,11,12]. Indeed, understanding the

genetics underlying disease susceptibility has become an important area of research; drugs with genetic support have been shown to be twice as likely to be successful in clinical trials[13,14].

Genome-wide association studies (GWAS) have identified common genetic risk variants at over 200 genetic loci associated with CRC risk, including those associated with anatomical subsite-specific CRC[10,15,16]. However, the mechanisms by which these genetic variants affect disease development are generally unknown, hindering

translation of these results into clinical applications. Most CRC genetic variants are located outside of coding sequences and their effects are assumed to be mediated through regulation of gene expression, adding complexity to the process of linking variants to the target gene. Given the potential to identify causal disease targets, establishing CRC susceptibility genes from GWAS presents an important opportunity for the development of new therapeutic targets. Indeed, studies have shown that genes or proteins identified through GWAS, or other genetic studies, of clinical phenotypes are more likely to be targeted by drugs approved for corresponding indications, compared to targets lacking such evidence[13,14].

Transcriptome-wide association studies (TWAS) are a form of post-GWAS analysis that establishes associations between gene expression and traits. In brief, gene expression is imputed to GWAS of traits of interest (here, CRC risk) using genetic variants which have been previously identified as being associated with gene expression in relevant tissues. Given the difficulty in accessing solid tissues for gene expression analyses, TWAS using these tissues are often limited by small sample sizes. S-MultiXcan and joint tissue imputation (JTI) are two TWAS methods which address this issue by incorporating information across multiple tissues to maximise statistical power[17,18]. Including multiple tissues in a single analysis also allows for the identification of the relevant biological tissue for the gene identified—which is important information for drug development. Notably, the S-MultiXcan approach also facilitates analysis of trait associations with alternative splicing events (i.e. processes producing distinct transcripts from the same gene). Alternative splicing is an often neglected mechanism in linking genes to traits despite evidence suggesting that up to ~30% of GWAS signals may mediate their effects through splicing[19].

TWAS have successfully identified potential susceptibility genes for many cancers, including breast[20], endometrial[21], and CRC[15,22–25]. However, no CRC TWAS performed thus far has stratified by anatomical subsite or sex, which are important aspects of CRC development[8,10,26]. Additionally, TWAS for CRC have often lacked a causal framework analysis to account for bias from residual linkage disequilibrium between genetic variants[15,25]. Consequently, it is likely that some previously identified genes represent spurious associations. Identifying genes that causally affect disease development is essential for revealing novel and effective avenues for CRC therapy and treatment.

In this study, we perform comprehensive multi-tissue expression and splicing TWAS analyses (outlined in Supplementary Fig. 1) to identify likely causal genes involved in CRC susceptibility, with a focus on sex- and anatomical subsite-specific associations. Here, we identify 37 genes with robust causal associations with CRC risk through a causal framework using Mendelian randomisation (MR) and genetic colocalisation. We highlight subsite-specific effects, such as rectal cancer risk linked to *LAMC1*, a clinically actionable drug target, and identify *CCM2* expression as a female-specific CRC risk factor involved in progesterone signalling. Our framework also prioritises *SEMA4D*, a previously unreported CRC susceptibility gene encoding a protein targeted by investigational cancer therapies. Additionally, we evaluate the impact of established drug targets on CRC risk by applying the same framework to 1163 genes encoding proteins targeted by approved or clinically studied drugs[27] and prioritise four such genes. Collectively, our findings provide important insights into the molecular mechanisms underlying CRC risk and reveal promising avenues for the development of new therapeutic strategies.

## Results

### Multi-tissue TWAS analyses
To identify genes associated with CRC risk at both the expression and splicing level, we used two multi-tissue TWAS methods: S-MultiXcan and JTI. For S-MultiXcan, we imputed gene expression using

expression quantitative trait loci (eQTLs) and splicing events using splicing quantitative trait loci (sQTLs). For JTI we imputed gene expression only as predictive models are not currently available for splicing events. For all TWAS approaches, gene expression or splicing events were imputed using data from the GTEx Project (version 8)[28]. We performed TWAS analyses using data from six tissues previously linked to CRC (subcutaneous and visceral adipose, lymphocytes, and whole blood) or directly relevant to CRC (sigmoid and transverse colon). Associations were tested with risk of overall CRC, as well as sex- or subsite-specific disease. CRC anatomical subsites were defined as per Huyghe et al.[10] (see "Methods"). Briefly, proximal, distal and rectal are mutually exclusive anatomical subsites designated by location of tumour, whereas colon is comprised of proximal colon and distal colon tumours, as well as colon cancer with unspecified location.

Across all three multi-tissue TWAS analyses, 112 unique genes were associated with CRC risk after Bonferroni correction ($p < 3.91 \times 10^{-7}$ in S-MultiXcan eQTL analysis; $p < 5.49 \times 10^{-7}$ in S-MultiXcan sQTL analysis; $p < 6.01 \times 10^{-8}$ in JTI analysis; Supplementary Fig. 2 and Supplementary Data 1–3). Of these genes, 64 were identified in the eQTL TWAS analyses, with 30 identified by both JTI and S-MultiXcan approaches. The splicing S-MultiXcan analysis revealed 144 unique splicing events associated with CRC risk, mapping to 60 genes, 23 of which were also identified in at least one of the eQTL TWAS analyses. None of the genes encoding proteins targeted by clinically studied drugs (i.e. 'druggable genes') passed correction for multiple testing in any of the TWAS analyses but 772 demonstrated nominal associations ($p < 0.05$).

### MR analyses
To evaluate the causal effect of gene expression on CRC risk, we performed MR, which uses germline genetic variants as instrumental variables to provide causal estimates (subject to certain assumptions, see Methods)[29,30]. Of the 112 genes identified by TWAS, 46 had available *cis*-genetic variants to proxy gene expression in at least one of the a priori selected tissues (minimum F-statistic: 30, median: 67). All genes had a single genetic instrument other than two genes (*MICA* and *MICB*), both of which had two genetic instruments. Among the genes with suitable genetic instruments, 29 passed multiple testing in MR analyses (Supplementary Data 4 and Supplementary Fig. 3). Of the 144 splicing events identified in the S-MultiXcan analysis, 37 had available genetic instruments to proxy the splicing event for MR analyses (minimum F statistic: 30, median: 63), with 27 passing the Bonferroni threshold, corresponding to 17 genes (Supplementary Data 5 and Supplementary Fig. 4). We also included the druggable genes in our causal framework analyses that were nominally associated with CRC risk from TWAS analysis, of which 380 had genetic instruments available according to our thresholds outlined in Methods (minimum F-statistic: 30, median: 60). The expression of seven of these genes passed multiple testing in MR analyses (Supplementary Data 6 and Supplementary Fig. 5).

### Colocalisation analyses
Genetic colocalisation analysis can help assess the evidence for causal associations between traits by evaluating whether the same or distinct variant(s) underlie the association between two traits[31]. Colocalisation analyses were performed based on the tissues identified in the TWAS: if a gene was identified in all six tissues in the TWAS, colocalisation analysis was performed for all six tissues. Conversely, if a gene was identified in only one tissue in the TWAS, colocalisation was restricted to that single tissue, and so on. Of the 112 genes identified by TWAS, there was evidence for a shared causal variant between gene expression for 29 of these genes and CRC risk ($H_4$, posterior probability of a shared causal variant between the traits, >0.80; Supplementary Data 7), and for 19 splicing events that mapped to 12 genes ($H_4 > 0.80$; Supplementary Data 8). Of the 29 genes prioritised by MR analyses, 20 had been prioritised by the colocalisation analysis; and of the

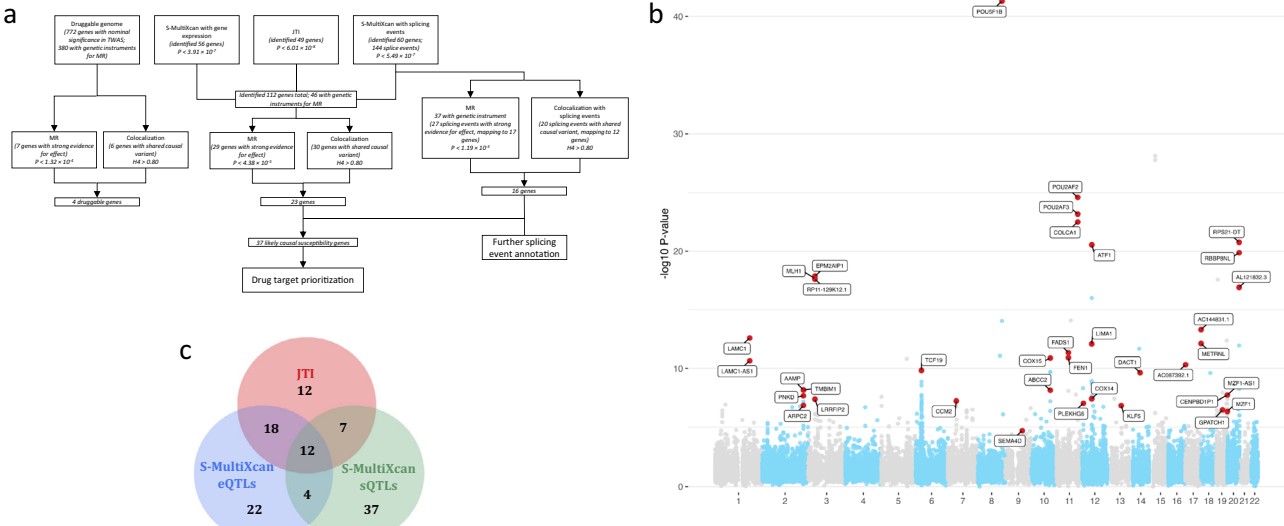

**Fig. 1 | Overview of multi-tissue TWAS, colocalization, and MR-based gene prioritisation for colorectal cancer risk. a** Flowchart showing analysis overview and number of genes/splicing events identified at each stage. "Genes with robust evidence" includes those that had $H_4$ above 0.8 in colocalisation analyses, and which either passed Bonferroni correction in the relevant MR analysis ($p < 4.38 \times 10^{-5}$; 0.05/$N*G$ where $N$ is the number of gene-tissue pairs (161) and $G$ is the number of CRC GWAS (7) for genes identified in TWAS analyses or $p < 1.32 \times 10^{-4}$; 0.05/number of druggable genes with suitable genetic instruments available (380) for genes identified as part of the druggable genome) or which did not have suitable instruments available to be included in the MR analysis. MR

Mendelian randomisation. **b** Manhattan plot showing results of S-MultiXcan and JTI TWAS analyses of colorectal cancer risk, for all anatomical subsites combined. Where genes were identified in multiple TWAS analyses, the one with the lowest $p$ value was retained. Genes labelled are those prioritised following subsequent analyses. All statistical tests were two-sided with the unadjusted $p$ values from S-MultiXcan or JTI plotted. **c** Venn diagram showing overlap of final prioritised 37 genes identified by each TWAS analysis. JTI joint tissue imputation, eQTLs expression quantitative trait loci, sQTLs splice quantitative trait loci. Source data are provided as a Source Data file.

27 splicing events prioritised by MR analyses, 12 were prioritised by the colocalisation analysis (corresponding to eight genes). Six druggable genes had evidence for a shared causal variant in colocalisation analyses ($H_4 > 0.80$) (Supplementary Data 9).

In order to avoid deprioritisation of CRC susceptibility genes or splicing events due to violations of the single causal variant assumption, we performed an additional colocalisation analysis using Pairwise Conditional Colocalisation (PWCoCo; described in "Methods")[32]. In brief, we applied PWCoCo to any gene or splicing event that met the multiple testing threshold in the MR analysis but had a $H_4$ posterior probability ≤0.80 in the standard colocalisation analyses. This resulted in the inclusion of an additional one gene based on expression (*TCF19*; Supplementary Data 10) and one splicing event (mapping to the gene *LRRFIP2;* Supplementary Data 11).

**Likely causal associations with colorectal cancer risk**

To identify likely causal gene associations with CRC risk, we used a stringent framework to prioritise genes: (1) passing Bonferroni correction in at least one TWAS analysis; (2) $H_4 > 0.80$ in genetic colocalisation analysis; and (3) passing Bonferroni correction in MR analysis or having no suitable genetic instruments available (Fig. 1a). Using this framework, we identified 37 genes with a likely causal association (Fig. 1b, Supplementary Fig. 6 and Table 1). Twenty likely causal susceptibility genes were identified solely through associations with expression and nine through associations with splicing alone. The largest magnitude of effect was observed for *POU5F1B* in the expression TWAS (*Z*-score in JTI = −13) and for *COLCA1* in the splicing TWAS (*Z*-score in S-MultiXcan with sQTLs = 10). We performed functional enrichment analysis of the likely causal genes using g:Profiler[33] and found significant enrichment ($p_{adj} < 0.05$) for genes involved in POU domain binding and the mitochondrial complex IV assembly (Supplementary Data 12).

Since we used two different methods for the expression TWAS (i.e. S-MultiXcan and JTI), we evaluated whether genes identified by

both methods were more likely to be prioritised by our framework (Fig. 1c and Supplementary Fig. 2). Of the 37 genes identified by both methods, 10 were prioritised (27%). In contrast, of the 19 gene expression associations identified by JTI alone, 12 were prioritised (63%), whereas only 2 of the 26 (8%) gene expression associations identified by S-MultiXcan were prioritised. These results suggest that JTI outperforms S-MultiXcan in prioritising genes with likely causal associations with CRC.

The likely causal genes included a previously unreported colorectal cancer susceptibility gene, *SEMA4D*, neither located at known colorectal cancer GWAS risk loci nor previously identified by colorectal cancer TWAS. A further ten genes were located at known colorectal cancer GWAS risk loci but had not been previously identified by colorectal cancer TWAS. Our analysis also revealed context-specific associations. Of the 37 likely causal genes, 23 showed tissue-specific associations (i.e. associations unique to expression or splicing in one tissue): five genes were found through analysis of subcutaneous adipose, one through visceral adipose, two through sigmoid colon, nine through transverse colon, three through lymphocytes and three through whole blood. Regarding anatomical subsites, two genes were exclusively associated with colon cancer risk (*AAMP* and *ARPC2*), three genes with both colon and proximal colon cancer risk (*EPM2AIP1*, *MLH1* and *RP11-129K12.1*), one with distal colon cancer risk (*ABCC2*), one with proximal colon cancer risk (*LRRFIP2*) and three with rectal cancer (*COLCA1*, *LAMC1* and *GPATCH1*) risk. For all but *AAMP*, differences in TWAS effect sizes for these genes were observed between subtypes (Figs. 2, 3). Lastly, one gene (*CCM2*) was specifically associated with female colorectal cancer risk (Fig. 2N).

For the analysis of the druggable genes, we conducted an exploratory analysis by focussing on genes that were nominally significant in at least one TWAS analysis. To prioritise genes for causality, we selected those passing $H_4 > 0.80$ in genetic colocalisation analysis and Bonferroni-correction in MR analysis. This approach revealed four

## Table 1 | Summary table of prioritised genes

| Loci | Gene | Ensembl ID | Direction | Tissue(s) | Subtype(s) | Analysis | Previous GWAS reference | Previous TWAS reference |
|---|---|---|---|---|---|---|---|---|
| 2q35 | AAMP | ENSG00000127837 | Increased | AS, AV, CS | C | eJTI | 16 | 15 |
| 10q24.2 | ABCC2 | ENSG00000023839 | Increased | WB | D | eMX | 16 | 15 |
| 17p13.3 | AC087392.1 | ENSG00000262003 | Decreased | AS, AV, CT | Overall, D, R | eJTI | 15,16 | None |
| 17q25.3 | AC144831.1 | ENSG00000261888 | Decreased | AS | Overall, R | eJTI, eMX | 16 | None |
| 20q13.33 | AL121832.3 | ENSG00000275437 | Decreased | CT | Overall, R, M | eJTI | 16 | 22,23 |
| 2q35 | ARPC2 | ENSG00000163466 | Increased | AS, AV | C | sMX | 16 | 15,22 |
| 12q13.12 | ATF1 | ENSG00000123268 | Increased | AS, AV, CS, CT | All analyses | eJTI, eMX, sMX | 16 | 15,22 |
| 7p13 | CCM2 | ENSG00000136280 | Decreased | WB | F | eJTI | 16 | 15 |
| 19q13.43 | CENPBD2P | ENSG00000213753 | Unknown | AS | Overall | sMX | 16 | None |
| 11q23.1 | COLCA1 | ENSG00000196167 | Decreased | CT | R | eJTI | 16 | 15,22,23 |
| 12q13.12 | COX14 | ENSG00000178449 | Increased | AS | Overall | eJTI | 16 | 15,22 |
| 10q24.2 | COX15 | ENSG00000014919 | Unknown | CT, L | Overall, D | sMX | 16 | 15 |
| 14q23.1 | DACT1 | ENSG00000165617 | Decreased | CT | C, P, M | eJTI, eMX | 16 | 15,22,23 |
| 3p22.2 | EPM2AIP1 | ENSG00000178567 | Increased | AV, WB | C, P | eJTI, sMX | 10 | None |
| 11q12.2 | FADS1 | ENSG00000149485 | Increased | AS, CS | Overall, C, D | eJTI, eMX, sMX | 16 | 15 |
| 11q12.2 | FEN1 | ENSG00000168496 | Increased | AS | Overall | eMX | 16 | 15 |
| 19q13.11 | GPATCH1 | ENSG00000076650 | Unknown | L | R | sMX | 16 | 15,22 |
| 2q35 | GPBAR1 | ENSG00000179921 | Increased | WB | Overall | Druggable genome | 16 | 15 |
| 13q22.1 | KLF5 | ENSG00000102554 | Unknown | CT | Overall | sMX | 86 | 15 |
| 1q25.3 | LAMC1 | ENSG00000135862 | Increased | WB | R | eJTI | 16 | 15,22 |
| 1q25.3 | LAMC1-AS1 | ENSG00000224468 | Increased | L | Overall, R | eJTI | 16 | None |
| 12q13.12 | LIMA1 | ENSG00000050405 | Decreased | CT | Overall, C, D | eJTI, eMX | 16 | 15 |
| 3p22.2 | LRRFIP2 | ENSG00000093167 | Unknown | AV | P | sMX | 10 | None |
| 12p13.31 | LTBR | ENSG00000111321 | Increased | AV, CS, CT | P | Druggable genome | 16 | 22 |
| 17q25.3 | METRNL | ENSG00000176845 | Increased | AV, WB | Overall, R, M | eJTI, sMX | 16 | 22 |
| 3p22.2 | MLH1 | ENSG00000076242 | Increased | AS, AV, L, WB | C, P | eJTI, eMX, sMX | 10 | None |
| 19q13.43 | MZF1 | ENSG00000099326 | Unknown | CS | Overall | sMX | 16 | 15,22 |
| 19q13.43 | MZF1-AS1 | ENSG00000267858 | Decreased | CS | Overall | sMX (MR expression) | 16 | None |
| 2q37.3 | PDCD1 | ENSG00000188389 | Increased | WB | R | Druggable genome | None | None |
| 12p13.31 | PLEKHG6 | ENSG00000008323 | Unknown | CT | Overall | sMX | 16 | 15 |
| 2q35 | PNKD | ENSG00000127838 | Increased | AV, WB | Overall, C | eJTI, eMX | 16 | 15,22 |
| 11q23.1 | POU2AF2 | ENSG00000150750 | Decreased | CT | Overall, C, R, D, M, F | eJTI, eMX | 16,88 | 15,22,23,88 |
| 11q23.1 | POU2AF3 | ENSG00000214290 | Decreased | CS, CT | Overall, C, R, D, M, F | eJTI, eMX, sMX | 16,88 | 15,22,23 |
| 8q24.21 | POU5F1B | ENSG00000212993 | Decreased | CT | Overall, C, P, F | eJTI, eMX | 16,88 | 15,23,88 |
| 1p31.1 | PTGER3 | ENSG00000050628 | Increased | AV | C, P | Druggable genome | 10 | None |
| 20q13.33 | RBBP8NL | ENSG00000130701 | Decreased | CT | C, F, M | eJTI, eMX | 16 | 22,23 |
| 3p22.2 | RP11-129K12.1 | ENSG00000272334 | Increased | AS, CT | C, P | eJTI, eMX | 10 | None |
| 20q13.33 | RPS21-DT | ENSG00000273619 | Increased | All tissues | C, F, M | eJTI | 16 | 15 |
| 9q22.2 | SEMA4D | ENSG00000187764 | Unknown | L | Overall, P | sMX | None | None |
| 6p21.33 | TCF19 | ENSG00000137310 | Increased | AS | Overall | eJTI, eMX | None | 15 |
| 2q35 | TMBIM1 | ENSG00000135926 | Increased | AS, AV, CS | Overall C | eJTI | 16 | 15,22,23 |

*Direction* whether the gene was associated with an increased or decreased risk of CRC, *Tissue(s)* the tissue(s) in which the gene was associated with CRC risk, *Subtype(s)* the CRC subtypes which the gene was associated with, *Analysis* the analysis (or analyses) in which the gene was identified, though note that for genes identified in the Splicing MultiXcan analysis the MR analysis, where carried out, evaluated evidence for a causal role of expression (not alternative splicing) of the gene in colorectal cancer risk, *Previous GWAS* any previously published CRC GWAS where this locus has been identified, *Previous TWAS* any previously published CRC GWAS where this locus has been identified. Tissue abbreviations: *AS* adipose subcutaneous, *AV* adipose visceral, *CS* colon sigmoid, *CT* colon traverse, *L* EBV-transformed lymphocytes, *WB* whole blood. Subtype abbreviations: *Overall* all CRC cases, *C* colon, *R* rectal, *P* proximal, *D* distal, *M* male, *F* female. Analysis abbreviations: *eJTI* expression JTI, *eMX* expression MultiXscan, *sMX* splicing MultiXcan.

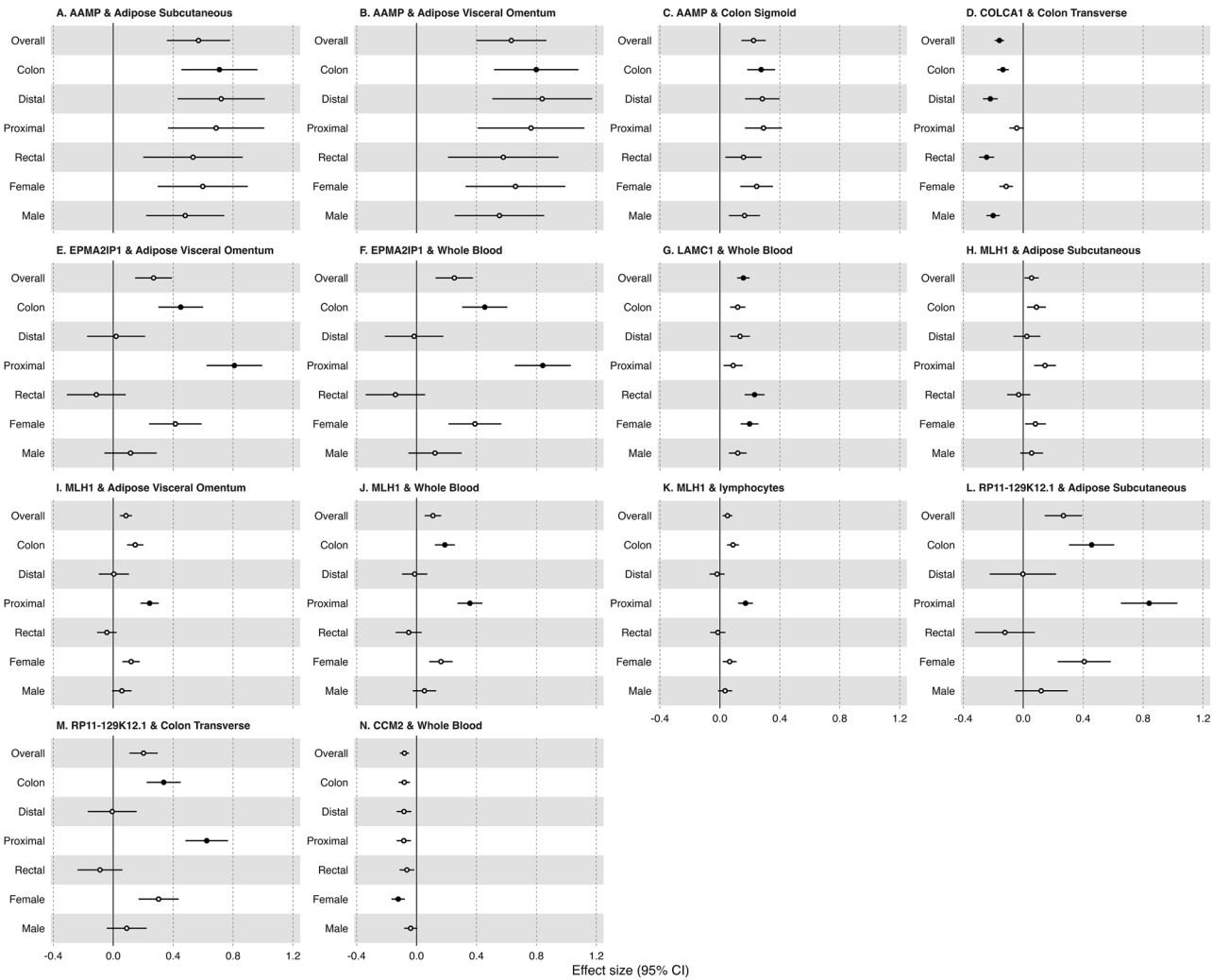

**Fig. 2 | Forest plots of JTI effect sizes across colorectal cancer anatomical subsites and sex for anatomical subsite- and sex-specific genes identified by JTI TWAS analysis.** Relevant tissue-specific estimates from JTI for risk of each anatomical subsite are plotted with 95% confident intervals. (sample sizes were 52,775 cases, 45,940 controls for overall; for all anatomical subsites there were 43,099 controls; colon, 28,736 cases; proximal colon, 14,416 cases; distal colon, 12,879 cases; and rectal, 14,150 cases; female, 24,594 cases, 23,936 controls; male, 28,271 cases, 22,351 controls). Solid points indicate the Bonferroni $p$ value threshold of $p < 6.01 \times 10^{-8}$ was met in the JTI analysis. Errors bars may be hidden by the point estimate where the standard deviation is small relative to effect estimates. **A** *AAMP* expression in adipose subcutaneous tissue; **B** *AAMP* expression in adipose visceral tissue; **C** *AAMP* expression in colon sigmoid tissue; **D** *COLCA1* expression in colon transverse tissue; **E** *EPM2AIP1* expression in adipose visceral tissue; **F** *EPM2AIP1* expression in whole blood; **G** *LAMC1* expression in whole blood; **H** *MLH1* expression in adipose subcutaneous tissue; **I** *MLH1* expression in adipose visceral tissue; **J** *MLH1* expression in whole blood; **K** *MLH1* expression in lymphocytes; **L** *RP11-129K12.1* expression in adipose subcutaneous expression; **M** *RP11-129K12.1* expression in colon transverse tissue; **N** *CCM2* expression in whole blood. Source data are provided as a Source Data file.

genes (*GPBAR1*, *LTBR*, *PDCD1* and *PTGER3*) (Fig. 1a, Supplementary Fig. 4 and Table 1).

### Splicing event annotation

To provide further support for likely causal splicing associations, we explored underlying splicing mechanisms. Using a bioinformatic splicing pipeline to analyse CRC GWAS risk variants for effects on the likely causal splicing events, we found that a single splicing event met the predetermined conditions indicative of a high-confidence splicing mechanism (see "Methods" for more information). This event, related to *PLEKHG6* (intron_12_6317696_6317899; Supplementary Data 13), could be explained by rs1468603 (chr12:6317886C > T). Specifically, the T allele was predicted to activate an exonic cryptic acceptor, enhancing the inclusion of a truncated exon 10 (45 bp in-frame deletion) in *PLEKHG6* (NM_001384598.1), corresponding to the intron_12_6317696_6317899 splicing event.

### Evaluating drug targeting opportunities provided by likely causal susceptibility genes

In addition to specifically analysing druggable targets, we investigated the druggability of proteins encoded by the likely causal susceptibility genes using the Pharos[34] and Open Targets[35] platforms to identify drug repurposing opportunities for preclinical or clinical investigation. These databases identified proteins encoded by *LAMC1* and *SEMA4D* as targets of clinically studied drugs. Laminin subunit gamma 1, encoded by *LAMC1*, is degraded by ocriplasmin, a recombinant proteinase drug used to treat vitreomacular adhesion. *SEMA4D* encodes semaphorin 4D which is inhibited by pepinemab, an antibody that has been clinically studied for treatment of several cancer types, including a phase I trial of CRC (Clinicaltrials.gov: NCT03373188). We also identified five genes (*ABCC2*, *ATF1*, *FADS1*, *FEN1* and *KLF5*) whose protein products bind to small molecules, supporting their potential druggability.

We evaluated the potential for efficacy in therapeutic targeting of likely causal susceptibility genes by assessing if their expression is

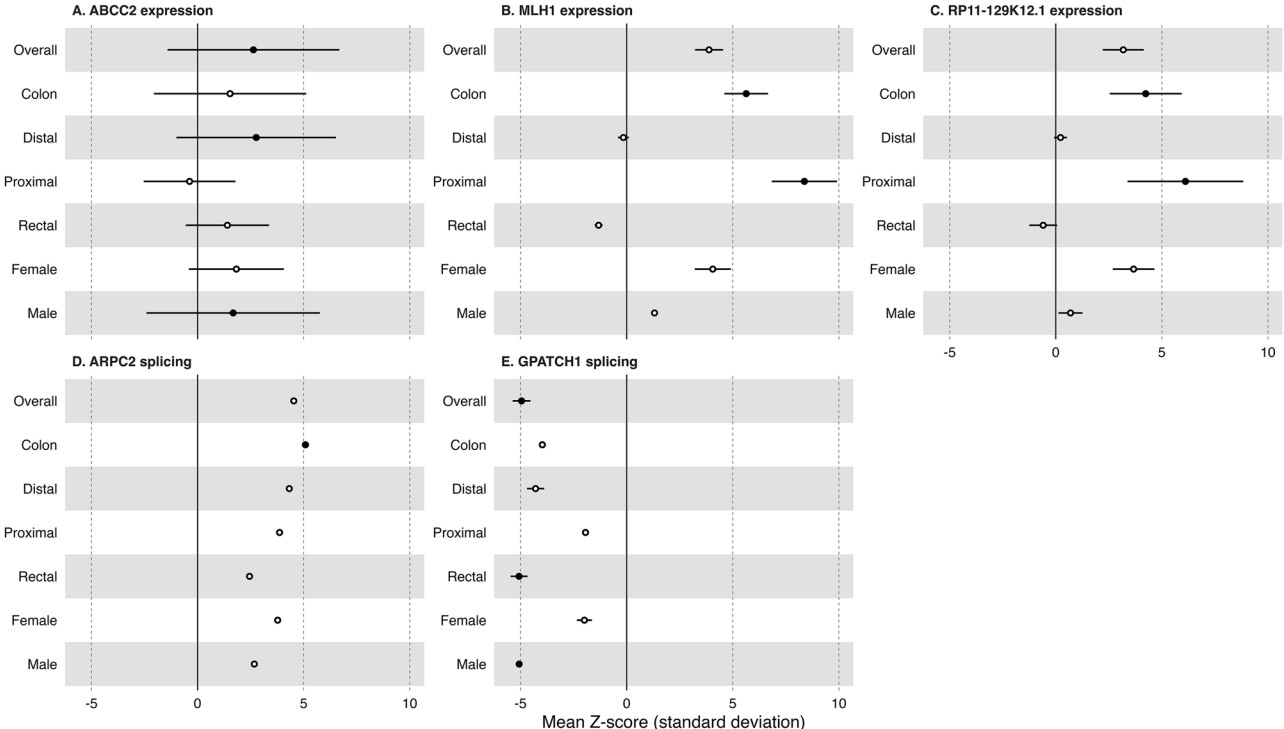

**Fig. 3 | Forest plots of mean *Z*-score estimates from S-MultiXcan across colorectal cancer anatomical subsites for anatomical subsite-specific genes identified by S-MultiXcan (expression or splicing) TWAS analysis.** Relevant estimates for risk of each anatomical subsite are plotted with 95% confident intervals (sample sizes were 52,775 cases, 45,940 controls for overall; for all anatomical subsites there were 43,099 controls; colon, 28,736 cases; proximal colon, 14,416 cases; distal colon, 12,879 cases; and rectal, 14,150 cases; female, 24,594 cases, 23,936 controls; male, 28,271 cases, 22,351 controls). Solid points indicate the Bonferroni *p* value threshold of $p < 3.91 \times 10^{-7}$ was met in S-MultiXcan eQTL analysis or $p < 5.49 \times 10^{-7}$ was met in S-MultiXcan sQTL analysis. Errors bars may be hidden by the point estimate where the standard deviation is small relative to *Z*-score scale. **A** *ABCC2* expression; **B** *MLH1* expression; **C** *RP11-129K12.1* expression; **D** *ARPC2* splicing; **E** *GPATCH1* splicing. Source data are provided as a Source Data file.

required for CRC cell line viability. Using the BioGRID Open Repository on CRISPR Screens[36], we found CRC cell lines were dependent on 16 of the likely causal susceptibility genes, with nine genes demonstrating dependency in at least 15% of studies with available data (Supplementary Data 14). Among these 16 genes, 11 were identified through expression TWAS approaches (Table 1). Consistent with the dependency findings, increased expression of eight genes, including *AAMP* and *FEN1*, associated with CRC risk. *AAMP* showed particularly consistent findings, with CRC cell lines demonstrating dependency for *AAMP* expression in 80% of the studies in which it was tested. CRC cell lines also showed frequent dependency for expression of *FEN1* (48% of studies), which encodes a potentially druggable protein.

### Shared causal pathways with known CRC risk factors

To investigate whether the likely causal susceptibility genes may relate to known CRC risk factors, we performed genetic colocalisation. We evaluated evidence for a shared causal variant between the expression of 28 likely causal susceptibility genes (i.e. those that passed both the colocalisation and MR thresholds, not including the seven genes that had robust evidence for splicing only) and each of four established CRC risk factors—BMI, WHR, alcohol consumption, and smoking initiation. Among these genes, we found evidence of colocalisation (posterior probability of $H_4 > 0.80$) for two genes (*AAMP* and *TMBIM1*) with WHR (Supplementary Data 15).

### Discussion

Our analysis combined two multi-tissue TWAS methods with a causal framework to identify CRC susceptibility genes. Through this framework, we prioritised 37 genes with strong evidence for a causal role in

colorectal cancer risk, with associations extending to specific disease subtypes and expression in distinct tissues, implicating the involvement of tissues outside the colon or rectum in CRC development. In addition, our analysis of the druggable genome revealed four genes with suggestive evidence for a causal role in colorectal cancer risk. The subsequent drug target analyses allowed us to highlight candidates for future investigation.

While previous TWAS for CRC have been conducted, these analyses have not been stratified by anatomical subsite or sex, which are important aspects of CRC aetiology. The importance of stratified analysis is demonstrated by our findings for a causal role of *CCM2* in female-specific colorectal cancer. Cerebral cavernous malformation 2 (CCM2) is a component of the CCM signalling complex, which has a role in regulating several signalling cascades, including progesterone signalling[37,38]. Notably, multiple studies have demonstrated a protective role for progesterone in CRC development (reviewed in Wenxuan et al.[39]). Our findings of decreased *CCM2* expression associating with increased CRC risk are consistent with this, supporting a potential sex-specific role for CCM2[37,38].

Nearly one third (11 of 37) of the susceptibility genes exhibited location-specific associations, highlighting the genetic heterogeneity of CRC. This subsite-level dissection provides a more nuanced understanding of this complex disease and underscores the importance of considering tumour location in genetic studies, with implications for developing more tailored treatment strategies. In addition, our findings are consistent with evidence from GWAS that genes at locus 3p22.2 (including *MLH1* and *EPM2AIP1*) have proximal colon cancer-specific effects[10,40,41]. Though loss of function *MLH1* variants are known to be associated with proximal colon cancer, we found that

increased *MLH1* expression was associated with increased cancer risk. A similar, albeit nominally significant TWAS finding was previously reported[22]. Supporting these observations, it has been reported that *MLH1* may have context-specific effects. For example, *MLH1* has been found to be upregulated in mismatch repair proficient CRC tumours and shown to have oncogenic effects in some contexts[42]. Nevertheless, further research is thus required to understand the direction of effect of *MLH1* expression on proximal colon cancer risk.

Among the likely causal genes, *SEMA4D* emerged as a CRC susceptibility gene that is neither located at known CRC GWAS risk loci nor previously identified by CRC TWAS. *SEMA4D* was identified through association of its alternative splicing with colorectal cancer risk, highlighting the importance of studying this mechanism using TWAS approaches. *SEMA4D* encodes a protein with immunoregulatory activity[43], consistent with its association with CRC risk through splicing effects in lymphocytes, also highlighting a potential causal cell type. Moreover, in a preclinical mouse colon cancer model, antibody blockade of SEMA4D has been shown to enhance the infiltration of immune cells into tumours, thereby promoting anti-tumour immune responses[44]. Importantly, our findings provide evidence to prioritise the clinical targeting of SEMA4D, currently being performed using an antibody treatment.

A further ten genes, located at known CRC GWAS risk loci had not been previously identified by CRC TWAS. These findings may possibly be due to the lack of anatomical subsite-stratified analyses in previous TWAS or our inclusion of alternative splicing events. Indeed, four of these genes (including *SEMA4D*) were exclusively identified through splicing associations. Further supporting the relevance of our splicing analysis, we demonstrated a potential mechanism for *PLEKHG6* splicing in CRC risk that involves the effect of a CRC GWAS SNP. These findings highlight the importance of incorporating splicing events in TWAS analyses, as they may reveal genes and mechanisms of genetic susceptibility that are not captured by gene expression alone.

*LAMC1* emerged as another likely causal susceptibility gene encoding a target of a clinically studied drug (ocriplasmin). *LAMC1* has previously been identified as a CRC susceptibility gene through GWAS and other approaches[15,45]. The laminin family of proteins are key components of the basal membrane and have been implicated in CRC progression[46,47]. We found genetically predicted increased expression of *LAMC1* was associated with increased rectal cancer risk, providing support for therapeutic inhibition of LAMC1. Ocriplasmin, a synthetic form of plasmin which targets laminin, is currently used to treat eye-related diseases and is also in phase II trials for several other conditions, including stroke and deep vein thrombosis[48–50]. While prior research has suggested ocriplasmin as a candidate drug for CRC treatment[51], further drug development would be required due to the current need for its direct injection and its moderate stability[52].

Evidence from publicly available data supports a role for several of the likely causal susceptibility genes in CRC, including *CCM2* and *SEMA4D* as discussed. Furthermore, mechanistic studies at the 11q23.1 CRC GWAS locus have linked risk variation to *POU2AF2* and demonstrated that this gene protects tuft cells in the colon while suppressing colonic tumourigenesis in a mouse model[53]. This observation is consistent with our TWAS finding that decreased *POU2AF2* expression is associated with increased CRC risk. Moreover, we have found that most likely causal susceptibility genes showing a dependency in CRC cell lines align with TWAS findings where increased expression was associated with increased CRC risk (e.g. *AAMP* and *FEN1*). This alignment underscores their relevance as candidate therapeutic targets. The most consistent findings of CRC dependency were for *AAMP* which encodes angio-associated migratory cell protein (AAMP), with a role in angiogenesis, cell migration[54], and CRC metastasis[55]. We also found evidence for colocalisation of *AAMP* expression with WHR suggesting that *AAMP* may also impact CRC risk through effects on adipose distribution, or vice versa. Although there are no current inhibitors of

AAMP, Open Targets indicates there is potential for inhibition through antibody or protein targeting chimera approaches. *FEN1* also demonstrated consistent CRC dependency. The metallonuclease encoded by *FEN1* has a role in DNA replication and double-strand break repair[56]. Promisingly, FEN1 small molecule inhibitors have been developed that show anti-cancer effects in experimental models[57]. These findings support the identification of druggable targets for CRC treatment, including corresponding candidate therapies or modalities, and provide valuable starting points for experimental validation and treatment development.

We also performed a comprehensive analysis of the "druggable genome"[27]. We focussed on genes that were nominally significant in at least one TWAS analysis and prioritised genes with evidence of genetic colocalisation ($H_4 > 0.80$) with CRC risk and which met the Bonferroni-correction in an MR analysis. This revealed suggestive evidence for a causal effect of expression of four genes (*PDCD1*, *GPBAR1*, *PTGER3* and *LTBR*) on CRC risk. Among these, there were two tissue-specific associations observed in whole blood (*GPBAR1* and *PTGER3*). Additionally, we found associations with unique anatomical subsite cancers: *LTBR* with risk of proximal colon cancer and *PDCD1* with risk of rectal cancer. *PDCD1* encodes programmed cell death 1 (PDCD-1 or PD-1) protein, which is targeted by inhibitors used to treat microsatellite instability-high or mismatch repair-deficient metastatic CRC[58–60]. Our TWAS and MR analyses suggested that increased (rather than decreased, replicating the use of an inhibitor) expression of *PDCD1* reduced risk of rectal cancer. This conflicts with evidence that PDCD-1 suppresses the immune system's ability to destroy cancer cells, as one would assume that in this case increased *PDCD1* expression would increase (not decrease) cancer risk[61]. However, we note that we only see strong evidence for a causal role of *PDCD1* expression in blood (not colon tissue) on cancer risk—suggesting that the mechanism linking *PDCD1* expression and colorectal cancer risk may be more complex than the presumed local effects within colorectal tissue. *PTGER3* encodes a receptor for prostaglandin E2 that is targeted by misoprostol, an approved drug for gastric ulcers and reflux disease and which has shown efficacy in colon cancer xenograft models[62]. We replicated previous GWAS evidence that *PTGER3* may have a role in proximal colon cancer and may be less relevant to rectal cancer[10]. *LTBR* encodes the tumour necrosis factor receptor lymphotoxin beta receptor (LTBR) which is targeted by an antibody agonist[63]. However, an antibody antagonist is likely to be required for effective treatment given increased *LTBR* expression in several tissues was associated with risk of proximal colon cancer.

Our analysis aimed to robustly prioritise genes for CRC susceptibility by using multiple tissues alongside a causal framework. We combined two genetic epidemiological approaches to assess genes spuriously identified due to linkage disequilibrium (i.e. showing evidence for a causal role in MR but not colocalisation) and to identify possible non-causal biomarkers of disease or risk factors (i.e. those that colocalise but show null results in MR analyses). However, the sample sizes for available data for TWAS analyses are still relatively small compared to the CRC GWAS, which potentially impacts our ability to genetically predict gene expression and detect associations with CRC risk. In addition, our analyses were limited to genes with expression that can be predicted using available TWAS models, meaning some potentially casual genes may not be captured in our analyses. Additionally, many of our MR analyses were restricted to a single SNP, meaning we were unable to employ various "pleiotropy-robust" models to evaluate exclusion restriction assumptions. We did not exclude *HLA* in the MR analyses, which is a possible limitation due to the region's high polymorphism and potential pleiotropic effects, which complicate causal interpretation. Linkage disequilibrium with other variants and unmeasured confounding factors further limit the ability to draw definitive conclusions. Furthermore, we did not evaluate the sensitivity of our colocalisation analyses to alternative window

sizes or prior probabilities, which are important aspects of colocalisation analyses[64]. Our study also presents further limitations that could be addressed in future research: (1) our analysed were restricted to individuals of predominantly European ancestries, which limits the generalisability of our findings to other populations and contexts; (2) the MR analyses performed here assume linearity between gene expression and CRC risk, which may not capture more complex interactions and non-linear relationships; (3) the use of available summary data limited our ability to perform analyses with sex-specific gene expression data that could provide insights into differential CRC risk; and (4) similarly, because we used summary-level data, we were unable to evaluate interactions between sex and CRC subtype.

Given the increase in CRC worldwide, understanding the biological mechanisms leading to carcinogenesis is becoming increasingly important[1]. Additionally, as more screening programmes are rolled out globally, opportunities to prevent CRC development in high-risk individuals are also increasing. Therefore, the identification of new pharmaceutical targets for the prevention and treatment of this disease remains a priority. Our analyses have identified genes with robust evidence for a potential causal role in CRC development, offering insights into its aetiology and presenting tangible opportunities for the exploration and development of new therapeutic strategies.

## Methods
### CRC GWAS
Supplementary Data 16 shows the GWAS used in all analyses. Summary genetic association data for CRC risk (52,775 cases, 45,940 controls) were obtained from a meta-analysis of the Colorectal Transdisciplinary Study (CORECT), the Colon Cancer Family Registry (CCFR), and the Genetics and Epidemiology of CRC (GECCO) consortium[10,16]. Summary genetic association data were obtained stratified by site (colon, 28,736 cases; proximal colon, 14,416 cases; distal colon, 12,879 cases; and rectal, 14,150 cases; 43,099 controls) and sex (female, 24,594 cases, 23,936 controls; male, 28,271 cases, 22,351 controls). Sex was defined based on sex chromosomes and samples with discrepancies between reported and genotypic sex based on X chromosome heterozygosity were excluded[10,16]. Colon cancer included proximal colon (any primary tumour arising in the caecum, ascending colon, hepatic flexure, or transverse colon), distal colon (any primary tumour arising in the splenic flexure, descending colon or sigmoid colon), and colon cases with unspecified site. Rectal cancer included any primary tumour arising in the rectum or rectosigmoid junction. CRC was classified using ICD-10 codes and most cases were incident CRC. All participants in the anatomical subsite-specific CRC analyses were of European ancestries, and approximately 92% of participants in the overall CRC GWAS were European (~8% were East Asian). Imputation of GWAS summary statistics was performed using the Michigan imputation server and HRC r1.0 reference panel. Regression models were adjusted for age, sex, genotyping platform, and genomic principal components as described previously[16]. All participants included in the CRC GWAS provided informed consent and ethics were approved by respective institutional review boards[10,16].

### Multi-tissue TWAS analyses
To identify genes with expression or splicing events associated with CRC risk, we utilised two multi-tissue TWAS methods. First, we performed S-MultiXcan[17], which is an extension of S-PrediXcan[65]. Briefly, S-PrediXcan identifies genes with expression or splicing events that are associated with a phenotype of interest using linear prediction models to impute gene expression and splicing events to the trait GWAS. We performed S-PrediXcan using precomputed gene expression or alternative splicing prediction models and linkage disequilibrium (LD) reference datasets of European ancestry, downloaded from the PredictDB data repository (http://predictdb.org/). S-MultiXcan extends this approach by incorporating gene expression prediction across

multiple tissues using multivariate regression. Effect sizes were calculated using multivariate adaptive shrinkage[66], which is a flexible statistical approach that leverages information on the similarity between variables to improve effect estimation. This approach was applied to variants identified by fine-mapping using deterministic approximation of posteriors[67,68], which performs joint enrichment analysis of GWAS and quantitative trait loci data to annotate genetic variants. Given that these models often rely on variants that may be absent from most trait GWAS, we performed additional harmonisation and imputation of the CRC GWAS prior to these analyses, as recommended by the S-MultiXcan authors. We performed the S-PrediXcan and S-MultiXcan analyses for both eQTLs and sQTLs. For the S-MultiXcan splicing analysis, splice events were mapped to relevant genes using the GTEx splicing mapping file (downloaded from www.gtexportal.org/home/datasets).

Second, we performed JTI as another means to identify genes with expression associated with CRC[18]. This method is another extension of S-PrediXcan and again imputes gene expression to trait GWAS by incorporating information across multiple tissues to improve prediction quality. We performed JTI using precomputed models for gene expression imputation which exploit measures of similarity between tissues based on expression data and cell-specific regulatory elements. The pretrained JTI models were downloaded from Zenodo (https://doi.org/10.5281/zenodo.3842289).

Both TWAS methods incorporate information about gene expression or splicing events across multiple biological tissues to maximise statistical power. As the architecture of eQTLs and sQTLs can differ substantially across tissues[28], previous evidence has suggested that using only those from tissues which are mechanistically related to the GWAS trait can avoid spurious findings[69]. Thus, for both TWAS methods, we used data (from GTEx Project version 8[28]) from six biologically relevant tissues for CRC: two adipose tissue types (subcutaneous adipose ($n = 581$) and visceral (omentum) adipose ($n = 469$)), which may capture important adiposity-related CRC pathways[2]; two colon tissue types (transverse colon ($n = 368$) and sigmoid colon ($n = 318$)), which may capture locally important oncogenic processes; one immune tissue type (Epstein-Barr virus-transformed lymphocytes ($n = 187$)), given recent links between circulating white blood cells and CRC risk[70]; and whole blood ($n = 670$), which may capture a range of clinically important circulating factors. We removed variants with a minor allele frequency (MAF) < 1% from the CRC GWAS summary statistics prior to TWAS analyses.

Given our aim of identifying genes which should be prioritised in future CRC research, for all TWAS analyses we applied a Bonferroni-correction to identify genes associated with CRC risk ($0.05/(N*G*T)$), where $N$ is the number of genes or splice events included in the analysis, $G$ is the number of CRC GWAS tested (overall, female, male, colon, distal, proximal, rectal), and $T$ is specific to the JTI analyses and is the number of tissues included in the analysis (of subcutaneous adipose tissue, visceral adipose tissue, transverse colon, sigmoid colon, lymphocytes, and whole blood). Any genes passing this Bonferroni threshold in at least one of the analyses ($p < 3.91 \times 10^{-7}$ in S-MultiXcan eQTL analysis; $p < 5.49 \times 10^{-7}$ in S-MultiXcan sQTL analysis; $p < 6.01 \times 10^{-8}$ in JTI analysis) were taken forward to the MR analyses.

S-MultiXcan aggregates expression predictions across multiple tissues to identify genes associated with CRC risk by leveraging shared genetic effects across tissues, which can increase statistical power. In contrast, JTI models gene expression across tissues while specifically accounting for tissue-specific effects, making it more sensitive to genes with distinct roles in particular tissues. Hence, S-MultiXcan and JTI may prioritise overlapping but distinct gene sets, with genes identified by both methods being more likely to represent robust associations.

Full S-PrediXcan results are available for download from Zenodo (https://doi.org/10.5281/zenodo.12805739).

## MR analyses

MR is a genetic epidemiological approach which, under certain assumptions, can estimate causal effects between phenotypes in observational settings[29,30]. MR uses germline genetic variants as instrumental variables for exposures. Since these variants are randomly assorted at meiosis and fixed at conception, MR analyses should be less prone to confounding by environmental factors and reverse causation bias than conventional observational studies. The three core assumptions of MR state that: (1) the genetic variant(s) are strongly and robustly associated with the exposure; (2) there is no confounding of the genetic variant(s)-outcome relationship (e.g., population stratification); (3) the genetic variant(s) only affect the outcome through their effect on the exposure.

We performed MR to evaluate evidence for a causal effect of tissue-specific gene expression for all genes identified in the TWAS analyses on the relevant CRC outcome (46 out of 112 genes were instrumentable). Summary genetic data for gene expression (i.e. eQTLs) were obtained from GTEx (version 8)[28]. We identified genetic instruments as genetic variants which are *cis*-acting (i.e. within 100 kb of the gene coding region), strongly associated with gene expression ($p < 5 \times 10^{-8}$), independent ($r^2 < 0.001$), and had an *F*-statistic >10. Steiger filtering was performed prior to MR analyses, with any genetic instruments explaining more variance in the outcome than the exposure excluded. See Supplementary Fig. 7 for an overview of our genetic instrument construction process. Where only a single genetic variant was available, we calculated the Wald ratio to generate effect estimates; where multiple genetic variants were available, an inverse variable weighted (IVW) multiplicative random effects model was used. A Bonferroni-correction was applied to account for multiple testing ($p < 4.38 \times 10^{-5}$; 0.05/N*G where N is the number of gene-tissue pairs (161) and G is the number of CRC GWAS (overall, male, female, colon, distal, proximal, rectal)). We additionally performed MR using sQTLs for splicing events, in order to assess their potential causal relationship with CRC outcomes, using the same thresholds for instrument construction and a Bonferroni-correction of $p < 1.19 \times 10^{-3}$ (0.05/42, the number of unique splicing event-tissue-subtype trios with suitable instruments for MR analyses).

In addition to genes identified through the TWAS analysis, given our focus on identifying genes which hold high therapeutic potential for CRC prevention, we also explored evidence for a causal role in CRC development of previously identified known druggable targets[27]. We limited genes included to those with nominal significance in at least one TWAS analysis, and we were able to identify genetic instruments to proxy the expression of 380 (out of 1163) of these genes for MR analyses. We used the same genetic instrument identification process as with the prior MR analysis and applied a Bonferroni correction to the results to account for multiple testing ($p < 1.32 \times 10^{-4}$; 0.05/number of druggable genes with suitable genetic instruments available (380)).

All genetic variants used in MR analyses are available in Supplementary Data 17. A completed STROBE-MR[71] checklist is available in as Supplementary Information (downloaded from: https://www.strobe-mr.org/).

## Colocalisation analyses

Genetic colocalisation uses a Bayesian framework to determine whether the causal variant(s) within a locus relating to multiple phenotypes is shared between the traits[31]. This shared causal variant is necessary (but not sufficient in the absence of other evidence) for a causal relationship. We performed genetic colocalisation under the single causal variant assumption[72] of (1) gene expression (eQTL) and CRC for all genes which were identified by any of the TWAS analyses and the relevant CRC anatomical subsite; (2) gene expression (eQTL) and CRC

for all genes from the aforementioned "druggable genome" for which data were available and all CRC anatomical subsites; and (3) gene splicing (sQTL) and CRC for all genes identified in the S-MultiXcan splicing analysis and the relevant CRC anatomical subsite. Colocalisation was performed using the priors $p1 = 1 \times 10^{-4}$, $p2 = 1 \times 10^{-4}$, and $p12 = 1 \times 10^{-5}$, with all genetic variants within 100 kb of the relevant gene coding region[72,73]. A posterior probability of >0.80 for $H_4$ was used to indicate strong evidence for a shared causal variant, and thus evidence for a causal relationship, between the traits.

In cases where the single causal variant assumption is violated and multiple variants at a given locus influence the trait, standard genetic colocalisation methods may produce false negatives. In our analyses, this could lead to the failure to prioritise a causal CRC susceptibility gene, particularly when strong evidence supports its role in CRC from TWAS and MR analyses but not genetic colocalisation. To assess whether our results were affected by violations of the single causal variant assumption, we performed an additional colocalisation analysis using Pairwise Conditional Colocalisation (PWCoCo)[32]. PWCoCo addresses the single causal variant assumption by performing iterative conditional colocalisation analysis. It first identifies the most strongly associated SNP at a locus. The association statistics for the remaining SNPs are then re-estimated while conditioning for the most strongly associated SNP, and the process is repeated iteratively until no further conditionally independent genome-wide significant ($p$ value $< 5 \times 10^{-8}$) signals remain. This approach therefore allows for the evaluation of multiple distinct causal variants for colocalisation between traits, rather than requiring that there is a single causal variant only. We applied PWCoCo to any gene or splicing event that met the multiple testing threshold in the MR analysis but had an H4 posterior probability ≤0.80 in the standard colocalisation analyses. PWCoCo was performed using all SNPs within ±100 kb of the gene coding region, with prior probabilities set at $p1 = p2 = 5 \times 10^{-5}$ and $p12 = 1 \times 10^{-6}$, selected based on the online calculator available at https://chr1swallace.shinyapps.io/coloc-priors/ (accessed 01/02/25).

## CRC dependency

To determine the dependency of CRC cell lines on likely causal susceptibility genes, we interrogated the BioGRID Open Repository of CRISPR Screens (https://orcs.thebiogrid.org/) and identified genes whose knockout impacts cell viability, using the study authors' defined threshold for evidence of gene dependency[36].

## Open Targets database

We used the Open Targets (https://www.targetvalidation.org)[35] and Pharos (https://pharos.nih.gov/)[34] platforms to evaluate drug target tractability and to identify drugs which may target the products of genes identified in our analysis.

## Splicing event annotation

We employed the SpliceAI-10k calculator to investigate downstream consequences of splice events, which has been described previously[74]. SpliceAI is a neural network trained on GENCODE-annotated pre-mRNA sequences and GTEx RNA-seq data to assess splicing variants for their likely splicing effects (i.e. loss or gain of acceptor or donor splice sites)[75]. The SpliceAI-10k calculator builds on this approach by using SpliceAI scores to systematically predict splicing aberrations (pseudoexonization, partial intron retention, partial exon deletion, exon skipping, and whole intron retention), altered transcript sizes, and consequent amino acid sequences[74]. In order to identify genetic variants to input to the SpliceAI-10k calculator, we performed fine-mapping using SuSiE[76] for all splicing events identified in the TWAS analysis using the relevant GTEx tissue splicing data with a window of ±100 kb around each splicing event. Genetic variants within credible sets were then filtered for those which were within 100:1 log likelihood of also being a CRC risk variant (i.e. genetic variant $p$ value is within two

orders of magnitude from the top genetic variant in the CRC GWAS). For splicing events for which no credible sets were identified, all genetic variants within 100:1 log likelihood of being a CRC risk variant were used. We then used the SpliceAI-10k calculator as previously described[77], to evaluate all resulting genetic variants for a high-confidence splicing mechanism based on whether they met three conditions: (1) they were predicted by the SpliceAI-10k calculator to impact splicing; (2) the predicted alternative exon matched with an Ensembl-annotated exon/transcript; and (3) this alternative transcript was the same as the alternative transcript identified in the original sQTL analysis in GTEx.

### Shared causal pathways with known CRC risk factors
To investigate shared causal pathways between our prioritised genes and known CRC risk factors, we performed genetic colocalisation as in our prior analysis. For each of the four previously identified CRC risk factors (BMI, WHR, alcohol consumption, and tobacco use), we performed colocalisation for expression of all genes with robust evidence (i.e. $p <$ Bonferroni threshold in relevant MR analysis and $H_4 > 0.8$ in colocalisation analysis) for a causal effect of expression on CRC risk, and the risk factor. We again applied a posterior probability threshold of $H_4 > 0.8$ as evidence for a shared causal variant between traits. In such cases, this suggests that there may be a shared causal pathway between expression of the gene and the risk factor. This could be indicative of a mediating role of expression of that gene in the effect of risk factors on CRC risk (e.g. increased BMI may increase expression of the gene which may increase risk of CRC). Alternatively, it may be that expression of the gene influences liability to the risk factor, which then increases risk of CRC through further biological pathways (e.g. if increased expression of the gene increases BMI, which then causes CRC through alternative pathways). We repeated analyses with sex-specific GWAS where data were available as a sensitivity analysis (i.e. for BMI and WHR; see Supplementary Data 16 for the sex-specific data sources).

### Statistical analyses
Units of gene expression betas, as outlined by the GTEx consortium, are the result of a normalisation procedure consisting of normalisation between samples using the trimmed mean of M values method[78], followed by normalisation across samples by inverse normal transformation, and as such the normalised expression units have no direct biological interpretation (see https://gtexportal.org/home/methods for more information)[28]. All analyses were performed using R version 4.0.2 or Python version 3.9.13 (other than the GWAS imputation step of the TWAS analysis which was performed using version 3.5.0). The following R packages were used: for colocalisation analyses, coloc[72] (version 5.1.0.1); for MR analyses TwoSampleMR[79,80] (version 0.5.5), gwasglue[81] (version 0.0.0.9000); for compiling LD reference panels, plinkbinr[82] (version 0.0.0.9000), ieugwasr[83] (version 0.1.5); for accessing Ensembl databases, biomaRt[84,85] (version 2.46.3); for finemapping, susieR[76,86] (version 0.12.35).

### Reporting summary
Further information on research design is available in the Nature Portfolio Reporting Summary linked to this article.

## Data availability
The data generated in this study can be found within the manuscript and supporting information, or the online repository on the Zenodo database under accession code 12805739. The CRC GWAS summary data used in this analysis are from GECCO (gecco@fredhutch.org; https://research.fredhutch.org/peters/en/genetics-and-epidemiology-of-colorectal-cancer-consortium.html). The CRC GWAS publicly available data underlying the summary statistics used in this study are

available in the dbGaP database under accession codes phs001415.v1.p1, phs001315.v1.p1, phs001078.v1.p1, phs001903.v1.p1, phs001856.v1.p1 and phs001045.v1.p1 (https://www.ncbi.nlm.nih.gov/gap/). Access to these restricted data can be requested through dbGaP. Other publicly available GWAS summary statistics used in this study are available in the GWAS Catalog under accession codes GCST006900 (body mass index), GCST002783 (body mass index−sex stratified), GCST008996 (waist:hip ratio), GCST008997 (waist:hip ratio−female), GCST008999 (waist:hip ratio−male), GCST007461 (alcohol consumption) and GCST007458 (smoking initiation) (https://www.ebi.ac.uk/gwas/home). The precomputed PrediXcan models were downloaded from http://predictdb.org and pretrained JTI model publicly available data used in this study are available from the Zenodo database under accession code 3842289. GTEx eQTL and sQTL summary data were downloaded from the GTEx website (https://www.gtexportal.org/home/downloads/adult-gtex/qtl). Source data are provided with this paper.

## Code availability
No custom code was generated for this study. All code is publicly available on GitHub at: https://github.com/EmmaHazelwood/CRC-TWAS-code and archived on Zenodo: (https://doi.org/10.5281/zenodo.12805738)[87].

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

## Acknowledgements

E.H. is supported by a Cancer Research UK Population Research Com-mittee Studentship (C18281/A30905), the CRUK Integrative Cancer Epidemiology Programme (C18281/A29019), the Harold Hyam Wingate Foundation, the European Cancer Prevention (ECP) organisation, the European Association for Cancer Research (EACR), and is part of the Medical Research Council Integrative Epidemiology Unit at the Uni-versity of Bristol which is supported by the Medical Research Council (MC_UU_00032/03) and the University of Bristol. T.A.O'.M. is supported by a National Health and Medical Research Council (NHMRC) of Australia Investigator Fellowship (Emerging Leadership 2; APP1179170). B.D. is supported by a Wellcome Trust studentship (218495/Z/19/Z) at the University of Bristol. D.M.C. is supported by NHMRC of Australia funding (APP177524). The CRC GWAS used in this analysis was provided by: GECCO, which acknowledges funding from the National Cancer Insti-tute, National Institutes of Health, U.S. Department of Health and Human Services (U01 CA137088, R01 CA059045, R01 201407), with genotyp-ing/sequencing services provided by the Center for Inherited Disease Research (CIDR) contract number HHSN268201700006I and HHSN268201200008I, and was funded in part through the NIH/NCI Cancer Center Support Grant P30 CA015704, with scientific Computing Infrastructure at Fred Hutch funded by ORIP grant S10OD028685; the CORECT study which was supported by the National Cancer Institute, National Institutes of Health (NCI/NIH), U.S. Department of Health and Human Services (grant numbers U19 CA148107, R01 CA081488, P30 CA014089, R01 CA197350; P01 CA196569; R01 CA201407; R01 CA242218), National Institutes of Environmental Health Sciences, National Institutes of Health (grant number T32 ES013678), and a gen-erous gift from Daniel and Maryann Fong; and the Colon Cancer Family Registry (CCFR, https://www.coloncfr.org/), which is supported in part by funding from the National Cancer Institute (NCI), National Institutes of Health (NIH) (award U01 CA167551), with support for case ascertainment provided in part from the Surveillance, Epidemiology, and End Results (SEER) Program and the following US state cancer registries: AZ, CO, MN, NC, NH; and by the Victoria Cancer Registry (Australia) and Ontario Cancer Registry (Canada). The CCFR Set-1 (Illumina 1M/1M-Duo) and Set-2 (Illumina Omni1-Quad) scans were supported by NIH awards U01 CA122839 and R01 CA143237. The CCFR Set-3 (Affymetrix Axiom COR-ECT Set array) was supported by NIH award U19 CA148107 and R01 CA81488 (to S.B.G.). The CCFR Set-4 (Illumina OncoArray 600 K genetic

variant array) was supported by NIH award U19 CA148107 (to S.B.G.) and by the Center for Inherited Disease Research (CIDR), which is funded by the NIH to the Johns Hopkins University, contract number HHSN268201200008I. Additional funding for the OFCCR/ARCTIC was through award GL201-043 from the Ontario Research Fund, award 112746 from the Canadian Institutes of Health Research, through a Cancer Risk Evaluation (CaRE) Program grant from the Canadian Cancer Society, and through generous support from the Ontario Ministry of Research and Innovation. The SFCCR Illumina HumanCytoSNP array was supported in part through NCI/NIH awards U01/U24 CA074794 and R01 CA076366. The content of this manuscript does not necessarily reflect the views or policies of the NCI, NIH or any of the collaborating centres in the Colon Cancer Family Registry (CCFR), nor does mention of trade names, commercial products, or organisations imply endorsement by the US Government, any cancer registry, or the CCFR. The funders had no role in study design, data collection and analysis, decision to publish, or preparation of the manuscript.

## Author contributions

E.H., D.M.G. and T.A.O'.M. conceived the initial study design. E.E.V., X.W., P.F.K., D.M.C., M.A.L. and D.N.L. contributed to the design of the study. E.H., D.M.C., B.D., and D.M.G. undertook all statistical and computational analyses. D.T.B., A.T.C., S.B.G., J.H., L.L.M., M.O.W., R.K.P., S.L.S., J.C.F. led the CRC summary genetic data acquisition, curation, and maintenance. E.H., D.M.G. and T.A.O'.M. drafted the manuscript. All authors (E.H., D.M.C., B.D., X.W., P.F.K., D.N.L., A.-E.C., M.A.L., D.T.B., A.T.C., S.B.G., J.H., L.L.M., M.O.W., R.K.P., S.L.S., J.C.F., W.Z., J.R.H., N.M., M.J.G., T.G.R., V.L.J.W., E.E.V., D.M.G., T.A.O'.M.) contributed to the interpretation of the results and critical revision of the manuscript. This work was supervised by T.A.O'.M.

## Competing interests

T.G.R. is employed full-time by GlaxoSmithKline outside of the research presented in this manuscript. Where authors are identified as personnel of the International Agency for Research on Cancer/World Health Organization, the authors alone are responsible for the views expressed in this article and they do not necessarily represent the decisions, policy, or views of the International Agency for Research on Cancer/World Health Organization. This article is the result of the scientific work of Dr Murphy while he was affiliated at IARC. The remaining authors declare no competing interests.

## Additional information

[1]MRC Integrative Epidemiology Unit, University of Bristol, Bristol, UK. [2]Population Health Sciences, Bristol Medical School, University of Bristol, Bristol, UK. [3]Population Health Program, QIMR Berghofer, Brisbane, QLD, Australia. [4]Cancer Research Program, QIMR Berghofer, Brisbane, QLD, Australia. [5]Division of Cardiovascular Medicine, Department of Medicine, Stanford University School of Medicine, Stanford, CA, USA. [6]Translational Health Sciences, Bristol Medical School, University of Bristol, Bristol, UK. [7]Nutrition and Metabolism, International Agency for Research on Cancer, WHO, Lyon, France. [8]Leeds Institute of Cancer and Pathology, University of Leeds, Leeds, UK. [9]Division of Gastroenterology, Massachusetts General Hospital and Harvard Medical School, Boston, MA, USA. [10]Channing Division of Network Medicine, Brigham and Women's Hospital and Harvard Medical School, Boston, MA, USA. [11]Clinical and Translational Epidemiology Unit, Massachusetts General Hospital and Harvard Medical School, Boston, MA, USA. [12]Broad Institute of Harvard and MIT, Cambridge, MA, USA. [13]Department of Epidemiology, Harvard T.H. Chan School of Public Health, Harvard University, Boston, MA, USA. [14]Department of Immunology and Infectious Diseases, Harvard T.H. Chan School of Public Health, Harvard University, Boston, MA, USA. [15]Department of Medical Oncology & Therapeutics Research and Center for Precision Medicine, City of Hope National Medical Center, Duarte, CA, USA. [16]Department of Medicine I, University Hospital Dresden, Technische Universität Dresden (TU Dresden), Dresden, Germany. [17]University of Hawaii Cancer Center, Honolulu, HI, USA. [18]Memorial University of Newfoundland, Discipline of Genetics, St. John's, NF, Canada. [19]Department of Pathology and Laboratory Medicine, Mayo Clinic, Arizona, Scottsdale, AZ, USA. [20]Genomic Medicine Institute, Cleveland Clinic, Cleveland, OH, USA. [21]Department of Molecular Medicine, Cleveland Clinic Lerner College of Medicine of Case Western Reserve University School of Medicine, Cleveland, OH, USA. [22]Department of Medicine, Samuel Oschin Comprehensive Cancer Institute, Cedars-Sinai Medical Center, Los Angeles, CA, USA. [23]Division of Epidemiology, Department of Medicine, Vanderbilt University Medical Center and Vanderbilt-Ingram Cancer Center, Nashville, TN, USA. [24]Public Health Sciences Division, Fred Hutchinson Cancer Center, Seattle, WA, USA. [25]Department of Epidemiology and Biostatistics, School of Public Health, Imperial College London, London, UK. [26]Conjoint Gastroenterology Laboratory, QIMR Berghofer Medical Research Institute, Herston, QLD, Australia. [27]Faculty of Medicine, The University of Queensland, Brisbane, QLD, Australia. [28]Conjoint Internal Medicine Laboratory, Pathology Queensland, Queensland Health, Brisbane, QLD, Australia. [29]These authors contributed equally: Dylan M. Glubb, Tracy A. O'Mara. ✉e-mail: Tracy.OMara@qimrb.edu.au

