## [Transparent Peer Review file · Nature Communications]

Multi-tissue expression and splicing data prioritise anatomical subsite- and sex-specific colorectal cancer susceptibility genes

Corresponding Author: Dr Tracy O'Mara

Version 0:

Reviewer comments:

Reviewer #1

(Remarks to the Author)

Enclosed is a review of Hazelwood et al's manuscript "Integrating multi-tissue expression and splicing data to prioritise anatomical subsite- and sex-specific colorectal cancer susceptibility genes with therapeutic potential." In this work, the authors conduct a gene- and splicing-event-aware TWAS analysis of colorectal cancer risk. The authors identified 10 novel genes, 35 total, and prioritize several interesting loci through follow-up analyses. Overall the paper is concise and written well and the methods are sound and well-motivated. I had a few overarching questions/comments:

My main comment revolves around the main premise of the paper, stemming from the title. The paper lacks adequate comparison between sex and subtypes and any interaction between the two. I would advise the authors to include more sex-specific analyses in their paper and highlights these in the results/figures.

In Lines 152-154, the authors write: "Of these genes, 68 were identified in the eQTL TWAS analyses, with 30 identified by both JTI and S-MultiXcan approaches." The number 68 does not agree with Figure 2.

Line 152: What is the P-value used for the Bonferroni correction here?

Why do the authors not conduct an MR test for the splicing events prioritized? There are even multivariate MR tests that can be employed here.

It could be helpful to compare/contrast the MR and colocalization results, showing overlaps and differences between the methods.

Line 240: where are the 28 prioritized genes from? The likely causal susceptibility genes among those 29 genes from colocalization results?

Reviewer #2

(Remarks to the Author)

Reviewer #3

(Remarks to the Author)

In this study Hazelwood and colleagues carry out TWAS and splicing analysis to identify colorectal cancer susceptibility genes that have potential therapeutic potential. I am not an expert on GWAS studies so can't comment on the appropriateness of the analysis but the results seem interesting and the authors identify a number of novel risk factors. Some of these show location or sex specificity and they propose that some of them could be therapeutic targets. Overall, however, it's not clear how much this study increases understanding of CRC risk beyond the addition of some additional potential risk genes. The findings quite restricted and seem quite modest compared to other recent studies (Law et al., 2024 for example). In addition, I think the authors overstate the importance of the findings and the potential for using this data for identifying therapeutic targets.

Specific comments below:

- 1) The authors carry out TWAS analysis to identify CRC susceptibility genes in different anatomical locations and across sexes. They identify a small number of genes that appear to demonstrate this but the relevance of these findings isn't clear. Mlh1 silencing/mutation for example, is known to be associated with proximal CRC and is identified as an overexpressed risk gene in this study also in proximal disease. This seems counterintuitive and no explanation is offered.
- 2) A similar comment regarding the drugability assessment. The authors identify some potentially druggable targets but it's unclear how useful this information is or what it could be used for.
- 3) Also, the dependency data is weak with most only being dependencies in very few lines. It's also not clear if these are also likely to be dependencies in normal tissue. Again, the usefulness of this information isn't clear.
- 4) The authors contend that the data provides evidence of the identified genes playing a causal role in CRC but no functional data is offered to support this. Can the authors investigate some of their novel hits in functional assays to provide evidence of this?

Together, I don't think the contention that the study identifies genes with robust evidence of causal role in CRC is supported by the data. It identifies potential candidates but much more work including functional evidence would be needed to support the authors conclusions. Overall, I am unconvinced that this study offers enough novelty and importance to warrant publication in Nature Communications.

Reviewer #5

(Remarks to the Author)

This study combines TWAS, mendelian randomization, colocalization, subsequent in silico drug target identification and validation analyzes to prioritize genes as drug targets for colorectal cancer (CRC). By doing this, it identified 35 genes among which four as drug targets for CRC. The methods are sound and there are multiple validation steps in support of the above findings. Generally, the paper is well written, with the exceptions of some improvements that are possible in the methodology (see major comments below).

Major reviews

Methodology:

- LCLs from GTEx are EBV-transformed lymphocyte cell types, altered by EBV virus. Some studies suggest this transformation affects gene expression, particularly genes influencing growth and survival (<https://journals.asm.org/doi/10.1128/jvi.00226-19>). Since these cells are immortalized and cancer-related genes are affected, does the exclusion of LCLs significantly alter results?
- L160 - Why wasn't MR performed for splicing events?
- Also, more information on the IVs is required: average F statistic and range, as well as number of significant MR findings with more than one SNP-IVs, and sensitivity analyses for pleiotropy when applicable.
- Given that splicing can independently affect phenotype and both eQTLs and sQTLs can be in LD, and considering shared genes between them, it's important to use SusiE plugin instead just mentioning this as a limitation.
- You should consider analyses like gene-enriched pathways
- In terms of choice of IVs for the MR, the pvalue threshold of 10^{-8} was applied but it seems to stringent for GTEx
- L516, If you want to evaluate if they have a mediating role, instead of performing a coloc for the risk factor, you could perform mediation analysis" → try the TwoStep MR framework ?
- An evaluation of the MR effect using Steiger Test should be tested.

Results section:

- An appreciation of the magnitude and direction of the effect of each causal gene on CRC is important, there should be somewhere a mention on the genes with the largest effects among the candidate gene list.

Minor reviews

Methodology:

- What is the main difference between S-MultiXcan and JTI? Since different genes are identified by each method, genes identified by both are likely more robust. Clarifying these differences could help us understand that these genes may not be false positives ?
- L162 - Of the 112 genes, only 46 had available cis-IV. Was it because they don't pass the p-value or F-statistic thresholds? This information was never mentioned.
- L387, reference datasets. European reference or Multi-Ethnic ?
- L532, too many packages cited (you should only cite packages used during the statistical analyses)
- Line 163: when it is mentioned cis-genetic variants, are these eQTL, sQTL, or either one or the other that were used as instruments in the MR analysis?
-

Results section:

- In the results section, the absence of association with sex or with specific subtype or CRC in the analyses should be highlighted.
- Coloc - Coloc analysis was performed on the 112 genes identified from TWAS, but only 29 passed multiple testing for MR. Are these the same 29 that colocalized? This is part is not clear in the results.
- Similarly in Coloc - 6 tissues were evaluated. Can you clarify if coloc was performed for all 6 or only under specific conditions, such as based on MR analysis results.
- L180 - You mention 3 criterias to prioritize genes. 1- Passing TWAS analysis; 2- Passing coloc; 3- Passing MR or having no IV. Figure 7 suggests only criterion 1 is necessary, followed by either criterion 2 or 3. Could you confirm this?
- L206 - Was this splicing event the only one showing significant results in this analysis?
- L455 - How did you define the list of genes with therapeutic potential? Was it from an online database? Was it only for drug targets linked to CRC or all known drugs?
- Table 1 : "ref ?" inside one case.
- If HLA was kept in the MR analyzes, the author should comment on the limitation of interpretation of the MR in the region.

Version 1:

Reviewer comments:

Reviewer #1

(Remarks to the Author)

I commend the authors for addressing my comments and recommend publication.

Reviewer #2

(Remarks to the Author)

Reviewer #3

(Remarks to the Author)

The authors have addressed my previous comments and I now recommend publication of this study.

Reviewer #5

(Remarks to the Author)

The authors have sufficiently addressed my previous comments. I have no further comments or suggestions.

REVIEWER COMMENTS

Reviewer #1 (Remarks to the Author):

Enclosed is a review of Hazelwood et al's manuscript "Integrating multi-tissue expression and splicing data to prioritise anatomical subsite- and sex-specific colorectal cancer susceptibility genes with therapeutic potential." In this work, the authors conduct a gene- and splicing-event-aware TWAS analysis of colorectal cancer risk. The authors identified 10 novel genes, 35 total, and prioritize several interesting loci through follow-up analyses. Overall the paper is concise and written well and the methods are sound and well-motivated. I had a few overarching questions/comments:

My main comment revolves around the main premise of the paper, stemming from the title. The paper lacks adequate comparison between sex and subtypes and any interaction between the two. I would advise the authors to include more sex-specific analyses in their paper and highlights these in the results/figures.

We thank the reviewer for their comments on the high quality and clear motivation of the submitted manuscript. We agree that further investigating sex- and subtype-specific effects of CRC, including interactions between the two, is an important area of research. However, because our analysis used summary-level data we are unable to evaluate interactions directly. We have now clarified this as a limitation of the paper in the Discussion section (lines 422-425). In addition, with the available data, we have made further comparisons of our subtype- and sex-specific results, including figures 2-3 to demonstrate the variable importance of each.

In Lines 152-154, the authors write: "Of these genes, 68 were identified in the eQTL TWAS analyses, with 30 identified by both JTI and S-MultiXcan approaches." The number 68 does not agree with Figure 2.

We apologise for this inconsistency and have now corrected this in the text (lines 159-160).

Line 152: What is the P-value used for the Bonferroni correction here?

We have clarified this in-text (lines 157-158).

Why do the authors not conduct an MR test for the splicing events prioritized? There are even multivariate MR tests that can be employed here.

Thank you for this suggestion. We have updated our paper to include MR analyses for splicing events, leading to a final list of 37 prioritized genes. We have described this additional analysis and the results throughout the manuscript.

It could be helpful to compare/contrast the MR and colocalization results, showing overlaps and differences between the methods.

We applied strict criteria to narrow down our TWAS-identified genes to prioritize the most likely causal susceptibility genes, requiring that genes must pass our predefined thresholds for both MR (if genetic instruments available) and genetic colocalization analyses. However, we agree that genes deprioritized by these thresholds may still provide important insights into the molecular underpinnings of CRC risk. We have clarified this in the Discussion section of the text (lines 401-405), providing context to the reader on what may be concluded from discrepancies between methods.

Line 240: where are the 28 prioritized genes from? The likely causal susceptibility genes among those 29 genes from colocalization results?

The genes raised by the reviewer are those which passed the colocalisation and MR thresholds - we have clarified this in the text (lines 285-287) and in Figure 1A.

Reviewer #2 (Remarks to the Author):

We recognize the contribution of the reviewer and thank them for their time.

Reviewer #3 (Remarks to the Author):

In this study Hazelwood and colleagues carry out TWAS and splicing analysis to identify colorectal cancer susceptibility genes that have potential therapeutic potential. I am not an expert on GWAS studies so can't comment on the appropriateness of the analysis but the results seem interesting and the authors identify a number of novel risk factors. Some of these show location or sex specificity and they propose that some of them could be therapeutic targets. Overall, however, it's not clear how much this study increases understanding of CRC risk beyond the addition of some additional potential risk genes. The findings are quite restricted and seem quite modest compared to other recent studies (Law et al., 2024 for example).

We appreciate the reviewer's comment regarding the modest number of novel risk genes identified in our study compared to other studies. However, it is important to note that studies that identify large numbers of risk genes typically use different and less stringent approaches to our study. Specifically, these studies do not prioritise risk genes based on causal relationships, such as we do using robust statistical genetic approaches. For example, Law et al. (2024) prioritised for functional GWAS variants rather than gene causality, relying on functional genomic data to infer potential regulation of target genes. Furthermore, they noted that '142 of the 208 candidate target genes...appear to have no documented role in CRC, and 47...presently have no established role in any cancer.', suggesting that a substantial proportion of these findings may be spurious. In contrast, our study uses causal inference methods (i.e. Mendelian randomization and colocalization), which emphasize specificity and yield fewer but statistically robust gene associations directly tied to CRC risk.

In addition, I think the authors overstate the importance of the findings and the potential for using this data for identifying therapeutic targets.

To avoid overstating the importance of findings, we have moderated the language regarding the potential for using these data to identify therapeutic targets throughout the Abstract, Introduction, Discussion and Conclusion. However, supporting our approach to use these data to identify potential therapeutic targets, it should be noted that drugs that target genes/proteins identified by genetic studies of clinical phenotypes are more likely to receive clinical approval than those whose targets lack such evidence. We have incorporated this information into the last sentence of the second paragraph of the Introduction (lines 104-107). As validation of our approach, one of our novel findings, *SEMA4D*, encodes a protein that is already being clinically targeted for treatment of colorectal cancer.

Specific comments below:

1) The authors carry out TWAS analysis to identify CRC susceptibility genes in different anatomical locations and across sexes. They identify a small number of genes that appear to demonstrate this but the relevance of these findings isn't clear.

We recognize that the number of susceptibility genes may initially appear modest but we believe these findings are highly relevant for the following reasons:

- 23 of the 37 prioritized CRC susceptibility genes have tissue-specific associations, providing evidence for the causal tissue. Not only do these findings suggest subsite specific CRC origins but also point to involvement of tissues or cell types outside the colon or rectum (e.g. subcutaneous adipose and lymphocytes) in mediating the effects of these genes. We have incorporated corresponding text into the first paragraph of the Discussion to highlight this (lines 295-296).
- Ten of the 37 genes prioritized for colorectal cancer susceptibility are novel findings that had not been previously identified by colorectal cancer TWAS (lines 333-334).
- The novel TWAS findings include *SEMA4D*, which not only appears to represent a new colorectal cancer GWAS risk locus but was also identified by a cell-specific splicing effect in lymphocytes, providing a potential causal cell type for this gene (we have added text to highlight this in the fourth paragraph of the Discussion, lines 323-332). As we now further discuss, *SEMA4D* encodes a protein that appears to inhibit immune cell infiltration and anti-tumour responses in a mouse model of colon cancer, supporting its relevance to CRC pathogenesis.
- Our study underscores the importance of incorporating splicing events in TWAS analyses, a key distinction from previous studies which focus on gene expression levels. Four of the novel TWAS findings (including *SEMA4D*) were exclusively identified from the splicing analysis. We also demonstrate the potential of splicing to uncover specific genetic susceptibility mechanisms that gene expression may not capture. For example, we demonstrate a splicing mechanism for the likely causal susceptibility gene *PLEKHG6* that involves a CRC GWAS SNP. We have highlighted these points in new text added to the fifth paragraph of the Discussion (lines 337-341).
- We present the first colorectal cancer TWAS analysis stratified by anatomical subsite, revealing that 11 of the 37 susceptibility genes exhibit subsite-specific associations. Four of these 11 genes had not been found by colorectal cancer TWAS, highlighting the importance of this analysis. Furthermore, by dissecting CRC susceptibility at this anatomical subsite level,

our analysis provides a more nuanced understanding of the genetic landscape of this complex disease, suggesting that CRC genetic risk factors can differ depending on tumor location within the colorectum, with implications for treatment strategies. We have added corresponding text to the third paragraph of the Discussion (lines 309-313).

- We prioritized *CCM2* for female-specific colorectal cancer susceptibility, the first sex-specific gene identified by CRC TWAS. The encoded protein is a component of a signaling complex, which has a role in regulating progesterone signaling. In the second paragraph of the Discussion, we now provide evidence for a protective role for progesterone in CRC development, highlighting our finding of decreased *CCM2* expression with CRC risk, which is consistent with this role (lines 303-308).

MLH1 silencing/mutation for example, is known to be associated with proximal CRC and is identified as an overexpressed risk gene in this study also in proximal disease. This seems counterintuitive and no explanation is offered.

We agree that our results suggesting that increased levels of *MLH1* increases cancer risk is surprising given the known relationship between silencing of this gene and proximal CRC development. However, it is possible that *MLH1* has context-specific effects in CRC, with a previous study demonstrating increased *MLH1* protein expression in MMR-proficient CRC tumours and oncogenic effects in some CRC models (PMID:21169277). We now provide this explanation in the third paragraph of the Discussion (lines 318-322).

2) A similar comment regarding the drugability assessment. The authors identify some potentially druggable targets but it's unclear how useful this information is or what it could be used for.

While we acknowledge that the direct therapeutic implications of identifying druggable targets may not be immediately apparent, we believe that this information is valuable. First, identifying targets with drugs clinically tested in CRC (e.g. pepinemab for *SEMA4D*) may help to prioritise ongoing investigations by highlighting targets that are causally linked to CRC. Meanwhile, targets with drugs approved for other indications provide opportunities for clinical drug repurposing. For preclinical studies, this approach also facilitates the selection of the corresponding druggable genes for experimental validation and provides initial drug candidates for development or modification (e.g. ocriplasmin for *LAMC1* and small molecule inhibitors for *FEN1*). To make our rationale explicit, we have added text to the corresponding section of the Results (lines 260-263) and to clarify implications of drugability, we have added a further sentence to the ends of the fourth and seventh paragraphs of the discussion (lines 331-332 and 371-374).

3) Also, the dependency data is weak with most only being dependencies in very few lines. It's also not clear if these are also likely to be dependencies in normal tissue. Again, the usefulness of this information isn't clear.

Although most dependencies are found in a minority of cell lines, 9 susceptibility genes demonstrated dependency in at least 15% of studies, as we now specify in the Results section (lines 272-275). Given the heterogeneity of CRC and that some CRISPR screens were conducted to model other factors (e.g. exposure to NK cells), it is likely that some dependencies reflect the diverse genetic/molecular profiles of CRC cell lines or are context-specific. Unfortunately, due to a lack of dependency data from normal

tissue, we are unable to assess their role in that context. Nevertheless, as we highlight in the Results section, among the 11 dependency genes identified through expression TWAS analysis, the increased expression of eight of these was associated with increased CRC risk, consistent with their dependency. As we had detailed in our Discussion, this consistency in findings coupled with the known biological functions of several of these genes (exemplified by *AAMP* and *FEN1*), underscore the relevance of these genes and support our approach. Furthermore, the findings for *AAMP* and *FEN1* highlight actionable targets for further study and therapeutic development. Notably, *FEN1* has already been identified as a target for small molecule inhibitors, which have shown anti-cancer effects in experimental models, validating its therapeutic potential.

4) The authors contend that the data provides evidence of the identified genes playing a causal role in CRC but no functional data is offered to support this. Can the authors investigate some of their novel hits in functional assays to provide evidence of this? Together, I don't think the contention that the study identifies genes with robust evidence of causal role in CRC is supported by the data. It identifies potential candidates but much more work including functional evidence would be needed to support the authors conclusions. Overall, I am unconvinced that this study offers enough novelty and importance to warrant publication in Nature Communications.

While our study emphasizes genetic and statistical evidence to prioritize genes for further research, we acknowledge that functional validation is a crucial next step to confirm causality in CRC. Although such studies are beyond the scope of the current project, as discussed in the manuscript (both previous and additional information covered in these responses), there are published studies that strongly support the causal roles of several of the genes identified in our study in CRC. For example:

- *SEMA4D*: This gene encodes a protein that has been shown to play a key role in immune regulation. In a preclinical mouse colon cancer model, antibody blockade of *SEMA4D* enhanced immune cell infiltration and promoted anti-tumor immune responses, providing evidence for a role in CRC pathogenesis.
- *CCM2*: We have highlighted the relationship between decreased *CCM2* expression and increased CRC risk, consistent with the protective role of progesterone in CRC development. This aligns with known biological pathways and supports *CCM2*'s involvement in CRC risk in females.
- *AAMP*: *AAMP* encodes the angio-associated migratory cell protein, which is involved in angiogenesis and cell migration. In vitro studies have shown that *AAMP* plays a key role in CRC metastasis. Moreover, the consistency of *AAMP* dependency in CRC cell lines (80% of studies) suggests its functional relevance in CRC development.
- *FEN1*: *FEN1* encodes a metallo-nuclease involved in DNA replication and double-strand break repair, crucial for maintaining genomic integrity. In CRC, *FEN1* has been shown to be essential for DNA repair and has been linked to CRC progression.

Furthermore, experimental evidence from Rajasekaran et al. (2024) demonstrate that *POU2AF2* protects tuft cells in the colon and functional analysis using CRISPR-mediated gene deletion in murine models show that loss of *POU2AF2* exacerbates tumorigenesis, consistent with our TWAS finding that decreased *POU2AF2* was associated with CRC risk. We have added corresponding text to the seventh paragraph of the Discussion (lines 354-358) to highlight this.

Reviewer #5 (Remarks to the Author):

This study combines TWAS, mendelian randomization, colocalization, subsequent in silico drug target identification and validation analyzes to prioritize genes as drug targets for colorectal cancer (CRC). By doing this, it identified 35 genes among which four as drug targets for CRC. The methods are sound and there are multiple validation steps in support of the above findings. Generally, the paper is well written, with the exceptions of some improvements that are possible in the methodology (see major comments below).

Major reviews

Methodology:

- LCLs from GTEx are EBV-transformed lymphocyte cell types, altered by EBV virus. Some studies suggest this transformation affects gene expression, particularly genes influencing growth and survival (<https://journals.asm.org/doi/10.1128/jvi.00226-19>). Since these cells are immortalized and cancer-related genes are affected, does the exclusion of LCLs significantly alter results?

Immunological traits, and in particular lymphocytes, are increasingly recognised as having an important role in CRC risk (DOI: 10.1002/ijc.34691). We thank the reviewer for raising the point that the gene expression data used is from lymphocytes which have been activated *ex vivo* with the EBV virus. We agree that this activation likely affects gene expression (as it presumably would also *in vivo*). However, we do not feel, based on current evidence, that there is any reason to believe that gene expression within activated lymphocytes is any less biologically relevant than in inactivated lymphocytes. Indeed, we are aware of ongoing work by our collaborators which makes use of lymphocyte data to examine the changing effects of gene expression on CRC risk over lymphocyte activation (using single-cell eQTLs identified in DOI: 10.1038/s41588-022-01066-3). In our TWAS results, we do not feel that our results are disproportionately driven by gene expression in lymphocytes (see Table 1 "Tissues" column). Therefore, we expect that the exclusion of this cell type would not meaningfully alter results.

- L160 - Why wasn't MR performed for splicing events?

In response to the reviewer's suggestion, we have performed MR on the prioritised splicing events, as detailed in the updated manuscript (see Results, MR analyses section starting line 166 and Methods, MR analyses section starting line 517).

- Also, more information on the IVs is required: average F statistic and range, as well as number of significant MR findings with more than one SNP-IVs, and sensitivity analyses for pleiotropy when applicable.

We have now included this information in the manuscript text (lines 169-172). We would like to note that given the proximal nature of gene expression to the genome and our use of *cis*-instruments to avoid pleiotropic pathways, only two genes had more than one genetic instrument available (*MICA* and *MICB*, both of which had two genetic instruments). Many commonly used pleiotropy-robust methods, for instance those which require a meaningful estimate of the mode or median genetic effects, are therefore not possible, but we note that our combination of MR with genetic colocalisation adds robustness to our results.

- Given that splicing can independently affect phenotype and both eQTLs and sQTLs can be in LD, and considering shared genes between them, it's important to use SusiE plugin instead just mentioning this as a limitation.

We thank the reviewer for raising this important issue and agree that our use of a colocalisation method with the single causal variant assumption could have led to important CRC susceptibility genes being missed. Therefore, we repeated analyses using the Pairwise Conditional Colocalization method (DOI: 10.1038/s41588-020-0682-6), which relaxes the single causal variant assumption similarly to the suggested use of coloc + SuSiE (see Results, Colocalisation analyses starting line 183 and Methods, Colocalisation analyses starting line 557).

- You should consider analyses like gene-enriched pathways

We have now performed this analysis and found enrichments for POU domain binding the mitochondrial complex IV assembly (Supplementary Table 12, lines 215-218).

- In terms of choice of IVs for the MR, the pvalue threshold of 10^{-8} was applied but it seems to stringent for GTex

The first assumption of MR stipulates that the genetic instrument must be “robustly” associated with the exposure of interest. In order to avoid false positives due to horizontal pleiotropy (i.e. inclusion of variants influencing an outcome through pathways independent to the exposure), we applied a threshold of $P < 5 \times 10^{-8}$, representing a Bonferroni correction for all genome-wide SNPs, as recommended in MR guidelines (DOIs: [10.12688/wellcomeopenres.15555.3](https://doi.org/10.12688/wellcomeopenres.15555.3) and [10.1016/j.sleep.2023.10.036](https://doi.org/10.1016/j.sleep.2023.10.036)). We feel that this stringent threshold is suitable for meeting our objectives of identifying and prioritising the most likely causal genes underlying CRC risk.

- L516, If you want to evaluate if they have a mediating role, instead of performing a coloc for the risk factor, you could perform mediation analysis” → try the TwoStep MR framework ?

We appreciate the reviewer’s suggestion to consider a mediation analysis using a two-step MR framework. However, we are currently unable to implement this approach given the unavailability of genome-wide summary genetic data for gene expression (e.g. GTEx data are only available for cis SNPs, or genome-wide significant trans SNPs). Specifically, this limitation prevents us from evaluating gene expression as an outcome (rather than as an exposure) in MR analyses, therefore meaning we are unable to complete the first step in two-step MR.

- An evaluation of the MR effect using Steiger Test should be tested.

We thank the reviewer for this suggestion. We have now conducted Steiger testing to confirm the directionality of analyses, as described in the revised manuscript (lines 532-534).

Results section:

- An appreciation of the magnitude and direction of the effect of each causal gene on CRC is important, there should be somewhere a mention on the genes with the largest effects among the candidate gene list.

We agree that the magnitude of effect, alongside the strength of evidence, is also important. We now include a brief description of the genes with largest effects among the candidate gene list in the Results (lines 213-215).

Minor reviews

Methodology:

- What is the main difference between S-MultiXcan and JTI? Since different genes are identified by each method, genes identified by both are likely more robust. Clarifying these differences could help us understand that these genes may not be false positives ?

We appreciate the reviewer's suggestion to clarify the methodological differences between S-MultiXcan and JTI. S-MultiXcan aggregates expression predictions across multiple tissues to identify genes associated with CRC risk by leveraging shared genetic effects across tissues, which can increase statistical power. In contrast, JTI (Joint Tissue Imputation) models gene expression across tissues while specifically accounting for tissue-specific effects, making it more sensitive to genes with distinct roles in particular tissues. Because of these differences, S-MultiXcan and JTI may prioritise overlapping but distinct gene sets. To enhance clarity, we have added a brief description of these differences in the Methodology section (lines 507-513). A comparison of genes identified before and after statistical prioritisation for causality (as shown in Supplementary Figure 2 and Figure 1C) reveals that 37% of the genes identified by both methods passed the statistical prioritisation for causality, whereas 63% of the genes identified by JTI and 8% of those identified by S-MultiXcan met these criteria. These results suggest that genes identified by both methods are only more robust in the case of S-MultiXcan and that JTI outperforms S-MultiXcan in prioritising genes with likely causal associations with CRC. We have added detail describing these findings to lines 219-226 in the Results section.

- L162 - Of the 112 genes, only 46 had available cis-IV. Was it because they don't pass the p-value or F-statistic thresholds? This information was never mentioned.

We thank the reviewer for pointing out the lack of detail surrounding the exclusion of genes for which no suitable genetic instruments were available. We now include a flowchart in our updated manuscript, which clearly shows the number of genes excluded with each step of instrument construction (Supplementary Figure 7).

- L387, reference datasets. European reference or Multi-Ethnic ?

We have now included this information (Lines 463).

- L532, too many packages cited (you should only cite packages used during the statistical analyses)

We have now updated this information accordingly (Lines 646-651).

- Line 163: when it is mentioned cis-genetic variants, are these eQTL, sQTL, or either one or the other that were used as instruments in the MR analysis?

This was either one or the other, depending on the analysis. This has been clarified in the manuscript (See Results, MR analyses line starting 166).

Results section:

- In the results section, the absence of association with sex or with specific subtype or CRC in the analyses should be highlighted.

Table 1 specifies which CRC phenotypes have demonstrated significant associations. We cannot confidently state that the lack of an observed association is due to there being no relationship between a gene and sex or specific CRC subtype, or if this is because of a lack of power.

- Coloc - Coloc analysis was performed on the 112 genes identified from TWAS, but only 29 passed multiple testing for MR. Are these the same 29 that colocalized? This is part is not clear in the results.

We have now clarified this information (See Results, Colocalisation analyses section line starting 183).

- Similarly in Coloc - 6 tissues were evaluated. Can you clarify if coloc was performed for all 6 or only under specific conditions, such as based on MR analysis results.

Coloc was performed on tissues based on results from the TWAS analyses. We have now clarified this information (Lines 186-189).

- L180 - You mention 3 criterias to prioritize genes. 1- Passing TWAS analysis; 2- Passing coloc; 3- Passing MR or having no IV. Figure 7 suggests only criterion 1 is necessary, followed by either criterion 2 or 3. Could you confirm this?

Genes were required to pass criterion 1, followed by both criteria 2 AND 3, as depicted by Figure 1A. If a gene could not be analysed by MR (i.e. did not have suitable genetic instruments available), then they only needed to pass criterion 1 followed by criterion 2.

- L206 - Was this splicing event the only one showing significant results in this analysis?

This is correct, and we have now clarified this in the text (Lines 249-252).

- L455 - How did you define the list of genes with therapeutic potential? Was it from an online database? Was it only for drug targets linked to CRC or all known drugs?

Genes with therapeutic potential were assessed using two databases (i.e. Pharos and Open Targets), as well as investigating if their expression is required for CRC cell line viability through the BioGRID Open Repository. We did not limit drug targets to those linked to CRC. Further information is found in

Results (Evaluating drug targeting opportunities provided by likely causal susceptibility genes line starting 258).

- Table 1 : “ref ?” inside one case.

Columns in Table 1 state whether the gene or locus has been reported by a previous GWAS or TWAS publication. If so, we have provided the citation. To clarify, we have modified the relevant column headers.

- If HLA was kept in the MR analyzes, the author should comment on the limitation of interpretation of the MR in the region.

We have now included this as a limitation (Lines 411-413).